# Developing programme theory for a place-based, systems change approach to adolescent mental health: A developmental realist evaluation

Kate Allen[1‡], Anna March[1‡], Bianca Alexandrescu[2], Julie Harris[3],
Rachael Stemp[4], Siying Li[2], Laura Kennedy[2], Karuna Davies[2], Tamanna Malhotra[1],
Ediane Santana de Lima[3], Tim Hobbs[3], Niran Rehill[5], Jenny Shand[2], Peter Fonagy[2],
Steve Pilling[2], Vashti Berry[1*]

1 Department of Public Health and Sport Sciences, Children and Young People's Mental Health Research Collaboration, University of Exeter, Exeter, United Kingdom, 2 Department of Clinical, Health and Educational Psychology, University College London, London, United Kingdom, 3 Dartington Service Design Lab, Buckfastleigh, United Kingdom, 4 Anna Freud Centre, London, United Kingdom, 5 University College London Partners, London, United Kingdom

‡ These authors are joint first authors on this work.
* v.berry@exeter.ac.uk

## Abstract

Kailo is a prevention framework that supports local communities to co-design evidence-informed strategies to address the social determinants of adolescent mental health and wellbeing. Similar place-based initiatives exist internationally but research into the mechanisms of change and contextual drivers is limited. Understanding how and why Kailo functions is crucial for refining the framework before it is implemented in different contexts and for informing the focus of an impact evaluation. We conducted a developmental, realist-informed evaluation to investigate how and why the Kailo framework functions, for whom and under what circumstances. The first phase of the evaluation involved a rapid realist review, observations of key processes, key informant interviews, and document analysis. These methods informed the development of a series of initial programme theories. The second phase involved testing and refining these initial programme theories through further interviews and focus groups with key informants. A realist dialogic approach was used throughout, and our work was supported by two Young People's Advisory Groups who provided input on both the approach and theory generation. We generated a series of programme theories that shed light on the underlying mechanisms through which the Kailo framework works, the ways in which context acts as a catalyst for change, and the outcomes prioritised. These programme theories fall into six themes: 1) Alignment; 2) Time; 3) Credibility; 4) Accessible Spaces; 5) Building a Shared Understanding/Mission; and 6) Creating Hope that Change is Possible. Our study has informed the further development of the Kailo framework and lays the groundwork for a subsequent contributory impact evaluation to assess how Kailo contributes to system-level change and,

**Data availability statement:** Relevant excerpts of the transcripts (interview and focus group) and observations are available within the maunscript. In compliance with our UCL Research Ethics Committee approval (UCL Ethics Number Project ID: 24683/001), full interview and focus group transcripts cannot be publicly shared, as participants did not consent to this and it would compromise participant confidentiality. All data availability requests can be sent to ethics@ucl.ac.uk.

**Funding:** This work is supported by the UK Prevention Research Partnership (MR/V049941/1), which is funded by the British Heart Foundation, Chief Scientist Office of the Scottish Government Health and Social Care Directorates, Engineering and Physical Sciences Research Council, Economic and Social Research Council, Health and Social Care Research and Development Division (Welsh Government), Medical Research Council, National Institute for Health Research, Natural Environment Research Council, Public Health Agency (Northern Ireland), The Health Foundation and Wellcome. This funding was awarded to VB, SP, PF and TH. In addition, the time of VB and PF is supported by the National Institute for Health and Care Research (NIHR) Applied Research Collaboration South West Peninsula and North Thames, respectively. The views expressed are those of the authors and not necessarily those of the National Health Service, the National Institute for Health and Care Research, or the Department of Health and Social Care. The funders had no role in study design, data collection and analysis, decision to publish, or preparation of the manuscript.

**Competing interests:** The authors have declared that no competing interests exist.

ultimately, improvements in adolescent mental health and wellbeing. The programme theories generated also contribute to a wider understanding of how complex place-based approaches may function.

## Introduction

Place-based approaches to improving mental health are increasingly prioritised over individual, standardised interventions, fostering community-level changes to enhance health outcomes [1–3]. These approaches acknowledge how the broader social determinants of health such as unemployment, job insecurity, racial discrimination, poverty, neighbourhood safety, transport, and housing insecurity [1,2,4] influence individuals' health behaviours and outcomes across the life course. They also recognise the complexity of how these determinants play out across different geographies [5,6], aiming to improve mental health and wellbeing by addressing local issues. These approaches take a variety of different forms, from local partnerships focusing on issue-based change to larger, government-led regeneration in a specific geographical area [6]. Place-based approaches usually focus on developing relationships in a community, and they are characterised by partnerships and shared design [7]. Examples of such approaches include youth mental health promotion and policy advocacy in Canada (Agenda Gap [8]), co-designed mental health support for 10–16 year olds in the UK (HeadStart [9]) and participatory action research to strengthen mental health services for Aboriginal and Torres Strait Islander young people in Australia [10]. Specific collaborators in place-based approaches will vary in relation to goals, but these initiatives all involve long-term work across multiple organisations or actors to create change [11,12]. Often these approaches act as catalysts for, or explicitly aim to achieve, wider systems change, influencing broader policies and practices and aiming to create population-level impacts [11,12].

Key components of place-based approaches tend to include effective multi-agency collaboration, engaging the local community in decision making, building a rich understanding of the local context, and ensuring local community autonomy [13,14]. For place-based approaches with ambitions for systems-level impact, early clarity on intentions for systems-level change has been hypothesised to be important as well as targeting multiple levels of prevention, investing in and sustaining collaborative partnership building efforts, and integrating data and evidence into strategy development [14]. Proposed benefits of such approaches include improved local-level relationships and collaboration, enhanced capacity, increased opportunity for co-creation, and greater effectiveness and sustainability [15].

Preliminary reviews have suggested place-based approaches may hold promise in reducing health inequalities and improving physical and broader health outcomes [16]. Specifically in relation to mental health and well-being, reviews indicate that context-specific approaches addressing social determinants and fostering collaborative working can positively impact quality of life, depression, and self-esteem [1], while also reducing risk factors and promoting protective factors against substance

misuse and violence in children and young people [17]. An example of this can be found in Communities that Care (CTC [18]). CTC is a predominately 'top-down' approach where system leaders commit to implementing a six-phase framework that provides the structure and tools to enable communities to engage with stakeholders, develop an understanding of local risk and protective factors, select pre-defined evidence-based strategies to address local needs, and implement and evaluate these strategies [18]. Not only has this approach resulted in increased use of evidence-based prevention strategies [19,20], but it has also been shown to reduce risk factors such as substance misuse and delinquency [18,21] and improve community protective factors [22].

Despite these promising findings, evidence indicates that initial positive impacts of place-based approaches are often not sustained [13] and, in some cases, their effects are isolated to a small number of individuals [23]. It is possible too, that a reliance on introducing pre-existing evidence-based programmes may lead to a lack of nuanced understanding of the priorities in a specific local context and miss opportunities to develop local ownership and innovation. As such, participatory approaches that elevate and integrate community voices at all stages of priority setting, design and development are required. Moreover, the evidence-base for place-based approaches remains limited, largely due to the challenges associated with evaluating the effectiveness of such complex programmes [24]. Place-based approaches inherently involve multiple stakeholders, are emergent by nature, operate within complex systems, and are intricately linked to local context while operating at multiple levels [24]. This complexity often renders traditional evaluation methods inadequate in capturing the ways in which place-based approaches are functioning [13]. Furthermore, these methods tell us little about the mechanisms underpinning these approaches and how they create change in local systems. *Developmental evaluations*, which involve real-time feedback to inform program adaptation and learning [25,26], have been highlighted as a valuable strategy for addressing some of these challenges [24]. Not only do they allow for continuous adaptation and responsiveness to emerging needs but also provide insights into how place-based initiatives might evolve and generate systemic change [25,26].

This study reports on a developmental evaluation [25,26] that begins to address this gap by generating knowledge about the mechanisms underpinning one specific place-based approach to improving mental health: *Kailo* [27]. Kailo is an adaptive, place-based framework targeting the social determinants of adolescent mental health and wellbeing, co-developed through developmental evaluation in two distinct UK communities [27]. The Kailo framework seeks to promote adolescent mental health and wellbeing by supporting local systems to develop locally relevant prevention strategies and address health inequalities, taking a bottom-up approach that innovatively integrates place-based practices with systems change (see Methods for more detail).

The early framework was developed by a consortium of academics, mental health practitioners, and social innovation and design agencies, with a theory of change grounded in principles of collaborative community change, systems modelling informed by scientific insight and local data, and participatory co-design practices [27–29]. Envisioned as a 3–5-year change process, Kailo is informed by ongoing evaluation with the aim of developing a scalable model.

## Aims of current study

Drawing on theory-driven realist research methods [30,31], this evaluation aimed to inform the development of Kailo and create a programme theory for its delivery in two pilot sites, including how and why it might lead to systemic change. Specifically, our study sought to address the following research questions:

1) How does Kailo function as an initiative? Why and for whom?

2) How is Kailo received in a local context and what conditions are necessary for place-based systems change to be achieved through Kailo?

3) What outcomes are prioritised in a place-based system through Kailo?

In order to provide sufficient detail on the different elements of the evaluation, our reporting is separated into two papers. This article focuses on the programme theory, while another describes how the evaluation fed into the developing Kailo framework and led to the creation of a detailed theory of change.

## Methods

### Ethics statement

The current study was approved by the Research Ethics Service, Office of the Vice-Provost (Research, Innovation & Global Engagement) at University College London (approval number: 24683/001). Formal informed written consent was obtained from all participants, including parents/guardians for each young people under 16 years of age who participated.

### Overview

To address our research questions, we undertook a realist-informed developmental evaluation [25,26,30–32]. Realist evaluation is a theory-driven approach that explores how causal mechanisms and contexts interact to produce outcomes (intended or unintended) [30]. This approach was chosen as it is particularly suited for evaluating frameworks like Kailo which are applied to complex systems that are emergent, dynamic, and variable in nature (see S1 Text for further rationale). We utilised multiple data sources to develop, test, and refine our initial programme theories (IPTs; see Box 1), which were constructed as a series of context-mechanism-outcome configurations (CMOCs; see Box 1). These data sources included a rapid realist review [33], review of Kailo programme documentation, observations of Kailo processes and activities, and interviews/focus groups with relevant Kailo informants (see Fig 1).

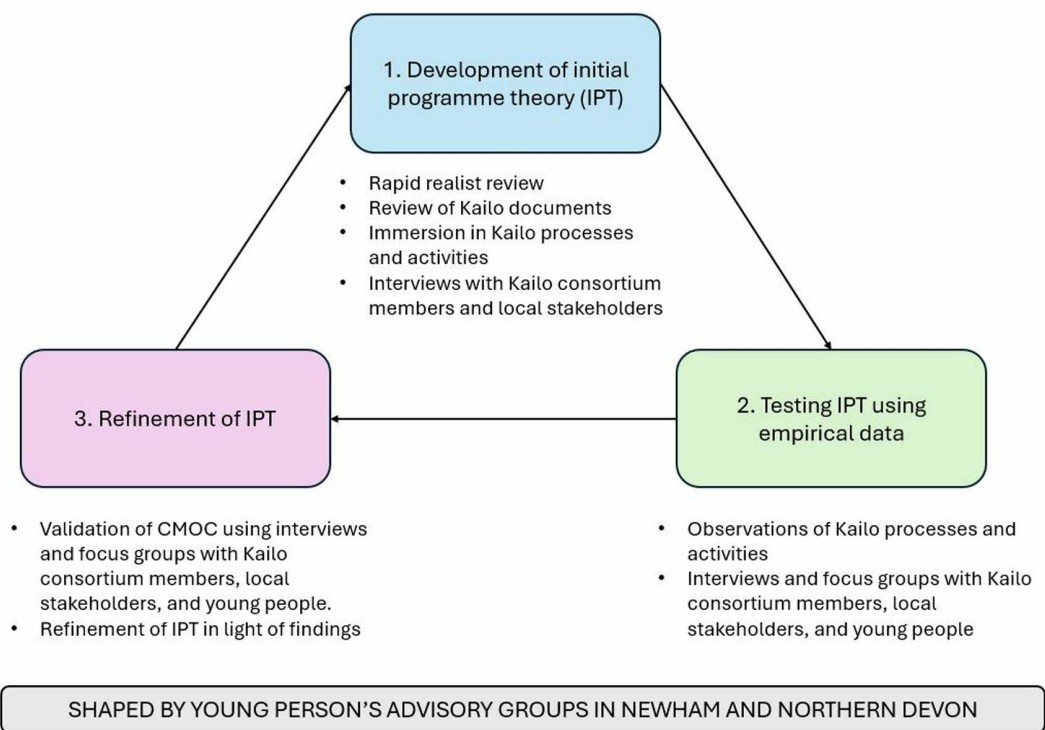

**Fig 1. Overview of Kailo developmental evaluation methods (adapted from Kennedy [ 35]).**

Our realist-informed developmental evaluation enabled us to address two critical objectives in the development and evaluation of Kailo (see [34]): (1) conducting a formative evaluation in the early stages of Kailo to build theory prior to an impact evaluation and (2) gaining a deeper focus and understanding of the role of context during initial delivery to support an effective scale-up of Kailo. Further information about our research design can be found in our protocol [35].

---

**Box 1. Definition of key realist terms (adapted from Jagosh, Bush [36])**

**Initial programme theory (IPT)** – Clear hypotheses about how, for whom, and in which contexts the Kailo framework might create change. In our developmental evaluation, IPTs comprise a series of CMOC statements (see below) and are developed, tested and refined as illustrated above and described below.

**Context-mechanism-outcome configuration (CMOC)** - Statements that explain observed outcomes. Creating a CMOC involves drawing out the relationship between context, mechanism and outcome in a particular programme (see S1 Text for further detail). In our developmental evaluation, CMOCs refer to specific aspects of the Kailo programme rather than the Kailo programme as a whole.

---

**The Kailo framework**

The Kailo framework (see [27] for more detail) has been developed in two pilot sites as part of the current evaluation – Newham and Northern Devon – and comprises three main phases:

1. *Phase 1: Early Discovery.* This phase involved activities to develop a detailed understanding of the local context. Key activities included fostering relationships and partnerships with local community stakeholders, identifying needs and opportunities, and forming communities around shared priorities (see Santana de Lima [37]) for an example of this process in Northern Devon).

2. *Phase 2: Deeper Discovery and Co-design.* This phase included co-design sessions (referred to as "small circle" sessions) with young people and community professionals to design solutions and strategies to address the priorities identified in Phase 1 (see Santana De Lima [38] for further details of this phase in both Newham and Northern Devon).

3. *Phase 3: Implementation phase.* This phase focused on the process of embedding and sustaining the strategies developed in Phase 2, ensuring their integration into the community.

These phases are guided by a set of key principles that emphasise *collaborative working with communities*, *building capabilities and adding value*, *recognising and working to reduce bias and inequalities*, and prioritising *ongoing reflection and learning*. Fig 2 provides an overview of the activities and funding for the phases of Kailo. S2 Text provides an overview of key Kailo terminology and stakeholders, while S1 Fig outlines the timing of Kailo phases in both sites alongside the timing of the evaluation activities. Table 1 details the phases of work undertaken in each site.

The current developmental evaluation focuses on the first two phases of Kailo: *Early Discovery* and *Deeper Discovery and Co-design*.

**Setting**

The current evaluation focuses on the Kailo framework implemented in two pilot sites: Newham and Northern Devon. While both of these pilot sites are based in England (UK), they differ significantly in terms of geographic and demographics offering a unique opportunity to assess how Kailo might function across different environments. Newham, an inner-city London borough, can be characterised as a densely populated, diverse community. In contrast, Northern Devon,

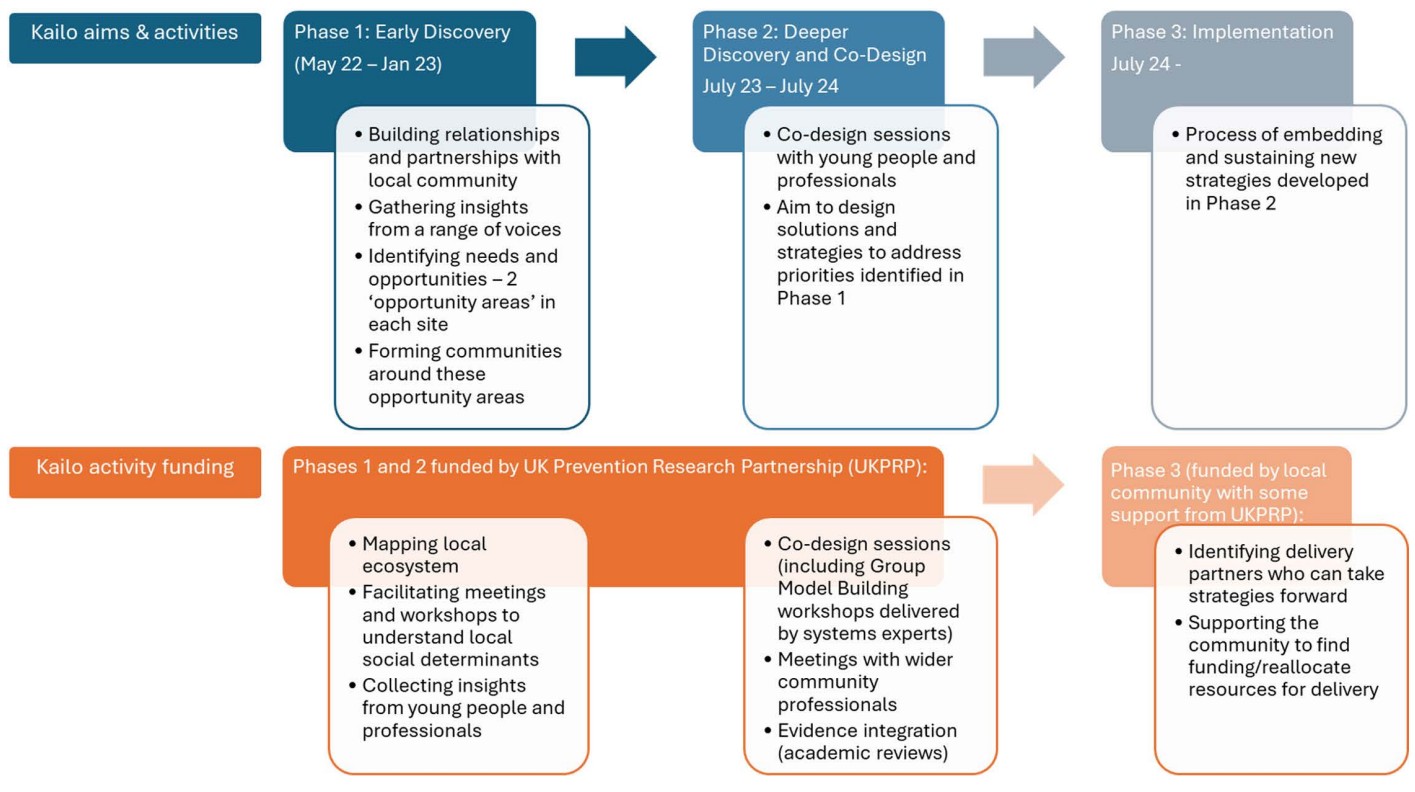

**Fig 2. Overview of Kailo activities and funding.**

a rural/coastal region in the Southwest, can be characterised as a sparsely populated, heterogenous community. Both Newham and Northern Devon have relatively high levels of deprivation [39]. The Kailo model places a particular emphasis on hearing from young people whose voices are often missing in conversations about the opportunity areas in each site. For the co-design sessions, the Kailo delivery team focused on recruiting young people in these underrepresented groups. This included neurodivergent young people, those identifying as LGBTQIA+, those living in isolated rural areas in Northern Devon and young people from racially minoritised groups in Newham. Co-design participants were recruited through community partner organisations who already had existing relationships with young people (e.g., a local charity providing support for autistic young people). In total, 44 young people aged 12–25 living in the Kailo pilot sites were recruited to participate in co-design sessions, incorporating diverse groups such as LGBTQIA+ young people, neurodivergent young people, those from white and mixed backgrounds, low-income households, and those from Muslim and Christian faiths.

### Data collection

**Rapid realist review.** To inform the development of Kailo's IPTs, the evaluation team conducted a rapid realist synthesis [40] of place-based approaches aimed at improving the mental health and wellbeing of children and young people. This review of international peer-reviewed and grey literature provided insights into the mechanisms underpinning place-based approaches similar in aim and scope to the Kailo programme. The findings were synthesised into 11 realist CMOCs across three overarching themes: (1) *building relationships and trust*, (2) *bringing a social determinants lens*, and (3) *educating and empowering community stakeholders*. These findings provided IPTs to be explored and tested during the observations, document analysis and interviews. Full details of the rapid realist review are available elsewhere [33].

**Table 1. Overview of Kailo phases in Newham and Northern Devon.**

| Sites | Early Discovery phase[1] | Deeper Discovery and Co-design phase[2] | | |
|---|---|---|---|---|
| | Opportunity area prioritised | Co-design sessions involved | Co-design sessions supported by | Strategies developed |
| **Newham** | | | | |
| Activities for wellbeing | Strengthening the role of local community infrastructure and activities for wellbeing | **Range per session:** 5–12 young people. **Total number engaged:** 12 young people. | • A local organisation focused on helping disadvantaged and at-risk children and young people.<br>• A local charity empowering the vulnerable in the local community. | Engagement strategy aiming to increase awareness and reduce stigma associated with youth clubs. |
| Violence and crime | Reducing the impact of violence and crime | **Range per session:** 5–11 young people. **Total number engaged:** 24 young people. | • A local youth-led organisation focused on preventing violence and promoting peace by empowering young people through education, sport, and personal development to reach their full potential.<br>• A local fashion and creative arts charity providing training and opportunities to young people.<br>• A local organisation providing support to young people affected by crime, gang-related activity, mental health issues, trauma and abuse. | 'Life Skills' centre – workshops for young people such as cooking classes, financial advice etc. |
| **Northern Devon** | | | | |
| Mental health literacy | Building stronger informal community support networks to promote mental health awareness and literacy, with a cross-cutting theme of fostering identity and belonging. | **Range per session:** 7–13 young people and 2–4 community professionals. **Total number engaged:** 15 young people and 7 community professionals. | • A large organisation supporting voluntary and community group development and capacity building.<br>• A Community Interest Company working with young people, adults, support providers and employers. | Strategies and materials to promote awareness of neurodiversity, LGBTQIA+, and mental health. |
| Diverse job opportunities | Creating and enhancing access to more diverse opportunities for studies, employment, and recreation, with a cross-cutting theme of fostering identity and belonging. | **Range per session:** 4–8 young people and 4–6 community professionals. **Total number engaged:** 15 young people and 11 community professionals. | •A local youth services organisation delivering professional youth work, community projects and other services to support young people. | A series of workshops to raise awareness of job opportunities in Northern Devon, as well as training to support employers to better engage with young people. |

NB Kailo activities in Newham were led by SHIFT in the *Early Discovery* phase and UCL Partners (https://uclpartners.com/) in the *Deeper Discovery and Co-design* phase. Kailo activities in Northern Devon were led by Dartington Service Design Lab (https://www.dartington.org.uk/) in the *Early Discovery* and *Deeper Discovery and Co-design* phases. 1See Santana de Lima, Preece [37] for further information about the *Early Discovery* phase. 2See Santana De Lima [38] for further information about the *Deeper Discovery and Co-design* phase.

**Programme documents.** Key programme documents elucidating the underlying rationale and Kailo processes were reviewed and analysed to inform the development of IPTs. These documents included governance and strategy documents (e.g., research bids and funder reports), internal reports, public-facing documents (e.g., blogs and recruitment materials) and academic papers. A total of 71 documents were purposively selected with guidance from senior Kailo consortium members responsible for co-ordination.

**Observations.** Initial observations of Kailo activities, processes, and meetings were undertaken during the *Early Discovery* phase to enable deep immersion in the programme's co-ordination and delivery. Further formal observations were conducted during the *Deeper Discovery and Co-design* phase to develop, test, and refine our IPTs. These included observations of delivery team meetings, co-design sessions, and community partner meetings. S3 Text provides a detailed description of these meetings/sessions.

**Interviews and focus groups.** Realist interviews and focus groups were conducted with Kailo consortium members, local community professionals, senior leadership, and young people in Newham and Northern Devon to help develop, test, and refine our IPTs. Data were collected in two rounds: **Round 1** (1st June – 30th November 2023) and **Round 2** (1st May – 30th June 2024). Table 2 provides an overview of the data collected.

**Table 2. Key informants sampled and data collection methods for rounds 1 and 2.**

| Key informant | Data collection method | Round 1 n | Round 2 n |
|---|---|---|---|
| **Kailo consortium members** | | | |
| Leadership/ co-ordination | Interviews | 5 | 2 |
| Newham | | 6 | 7 |
| Northern Devon | | 2 | 5 |
| **Local community professionals and senior leadership** | | | |
| Newham | Interviews | 2 | 5 |
| Northern Devon | | 3 | 11 |
| **Young people** | | | |
| Newham | Focus groups and interviews | – | 1 FG; 4 YP 3 INTs |
| Northern Devon | | – | 3 FGs;12 YP |

FGs = focus groups; INTs = interviews; YP = young people

A total of 18 interviews were conducted in *Round 1* to aid the development of our IPTs, while 30 interviews and four focus groups were conducted in *Round 2* to test and refine these IPTs. We purposefully sampled informants based on their detailed knowledge of programme implementation and how Kailo had been received in Newham and Northern Devon (*Round 1*) or their involvement in leading, delivering, or engaging with the *Deeper Discovery and Co-design* phase of Kailo (*Round 2*). Some informants were recommended to us by the Kailo delivery team.

All interviews and focus groups were conducted either online (via Microsoft Teams or Zoom) or in person by a member of the evaluation team, using interview schedules informed by realist principles. *Round 1* interview schedules included exploratory questions designed to establish how Kailo operates, for whom, and in what circumstances, focusing on exploring specific contexts, mechanisms, and outcomes identified as potentially important (see S4 Text). *Round 2* schedules were informant-specific and focused on testing and refining IPTs. These schedules followed the 'learner-teacher cycle' model, wherein members of the evaluation team presented theories and ideas to informants to challenge, debate, and refine [41,42]. Schedules were adapted to informants' prior understanding of programme theories (see S5 Text). There was some fluctuation in attendance across the co-design sessions in both Northern Devon and Newham. Despite efforts from the organisers to fit sessions around young people, some found that sessions clashed with their work or school commitments, while others did not provide a reason for not attending. As per our ethical approval, the Kailo site teams had responsibility for communicating with young people and, after reasonable efforts were made to re-engage those who stopped attending the sessions, we treated non-response as an opt-out of the research.

Interviews and focus groups lasted an average of 47 minutes and were audio-recorded and transcribed verbatim for analysis.

## Data analysis

**Developing IPTs.** IPTs were developed through the rapid realist review, immersion in Kailo processes and activities, analysis of Kailo documents, and Round 1 of interviews with informants. In parallel, the evaluation team immersed themselves in Kailo activities and processes, informally observing Kailo meetings, reading and re-reading Kailo documents, and regularly meeting with Kailo consortium members to discuss the ongoing development of the framework. This process enabled the evaluation team to identify important contexts, mechanisms, and outcomes to explore further within key documents and interviews. Data from key documents and interviews were line-by-line coded using these contexts, mechanisms, and outcomes as codes.

Weekly meetings among the evaluation team facilitated discussion and reflections on the analysis, using a realist dialogic approach [43] to collaboratively develop an initial series of CMOCs that attempted to explain how Kailo was functioning, for whom, and under what circumstances. This first stage of the analysis resulted 31 CMOCs (see S6 Text).

**Testing and refining IPTs.** IPTs were then tested and refined using data from observations and Round 2 of interviews and focus groups. Observation and interview/focus group data were coded against the initial 31 CMOCs. Researchers took detailed reflexive notes on where the data supported or refuted the CMOCs, and identified areas for adaptation.

The evaluation team continued to meet weekly, using a realist dialogic approach [43] to analyse the data. The analysis was finalised following a series of in-depth sessions to integrate collective learning and refine the initial CMOCs. This second stage of the analysis resulted in a final set of 12 CMOCs which describe how Kailo operates, for whom, and under what conditions (see Results).

### Young person's advisory groups

The developmental evaluation was guided by two young person's advisory groups (YPAGs): one in Newham and one in Northern Devon. These groups included young people aged 16–25 who were living in the local area and were independent from the young people involved in co-design sessions. The YPAGs met monthly between September 2023 and August 2024 in a mix of online and in person sessions. The groups spent time at the beginning getting to know each other, as well as the research methodologies. The groups were designed as a safe space to share ideas and opinions, with flexible activities and options for engagement to ensure that all young people could contribute meaningfully and creatively. The YPAGs undertook several roles, including rewording participant information leaflets and consent forms, reviewing interview/focus group schedules for Round 2, and revising CMOCs to ensure they were appropriate and accessible for young people. The YPAGs also created lay summaries of the developmental evaluation and co-facilitated focus group discussions with other young people.

Both YPAGs were led by RS and sessions were designed following the principles outlined in the Lundy model of child participation [44], ensuring meaningful engagement and respect for young people's perspectives.

## Results

The developmental realist evaluation of Kailo resulted in 12 CMOCs that explain the mechanisms through which Kailo was operating in Newham and Northern Devon (**RQ1**), the context and conditions necessary for systems change (**RQ2**), and the outcomes prioritised through Kailo's place-based approach (**RQ3**). We organised these CMOCs into six categories including: (1) **Alignment**; (2) **Time**; (3) **Credibility**; (4) **Accessible Spaces**; (5) **Building a Shared Understanding/Mission**; and (6) **Creating Hope that Change is Possible**. An overview of these themes and their associated CMOCs is provided in Table 3. We detail how the CMOCs address each of our three research questions in the discussion.

### 1. Alignment

#### 1.1. Alignment with stakeholder and community values, priorities, and needs

Young people, community professionals, and system leaders talked about how Kailo's methods or mission aligned with their personal or community values (beliefs, principles, standards), priorities, and needs. This alignment was key to ensuring the community felt enthusiastic about the work, being interested in engaging and continuing to support Kailo throughout the *Early Discovery*, *Deeper Discovery and Co-design*, and even *Implementation* phases. As such, it was essential that Kailo consistently highlighted the ways in which it aligned with these values, priorities, and needs.

**Young person level.** Young people described wanting to take part in Kailo because the mission aligned with personal values and priorities to contribute to better supporting the community they lived in. This was evident in both Northern Devon and Newham.

**Table 3. Themes and CMOCs developed as part of the developmental evaluation.**

| Theme | CMOC |
|---|---|
| **1. Alignment** | 1.1: Where stakeholders or the broader community are looking for something (C1) and the programme team have developed an understanding of this (C2), demonstrating alignment with the community's needs, priorities, or values (MResO) means stakeholders feel enthusiastic about the goals of the programme (MResP). This means stakeholders are motivated to engage in the work (O1), committed to sustained involvement (O2), and champion the work (O3). |
| | 1.2: Where community professionals recognise an alignment between programme activities and their daily work (C), the programme team modelling how to effectively involve, empower, and learn from young people (MResO) motivates community professionals (MResP). This results in community professionals applying their learning to their daily work/own organisations (O). |
| **2. Time** | 2.1: Where there is time dedicated to programmes (C1) and programme teams are consistently present in the community (C2), actively listening to the communities' views and concerns (MResO), demonstrates commitment and facilitates the building of trusting relationships (MResP). This leads to a better understanding of the community, community context, and relevant existing initiatives (O1), results in recruiting the right people to be part of the work (O2), and sets the foundation for collaborative partnership working (O3). |
| | 2.2: Where programmes have time (C1) and programme teams are open to learning and adapting (C2), demonstrating flexibility by continuously understanding, adapting and responding to the needs of community stakeholders (MResO) means community stakeholders feel heard (MResP). This leads to community stakeholders feeling valued and willing to participate (O1) and generates local insights into the social determinants of adolescent MH (O2), enabling the programme to respond to community need (O3). |
| **3. Credibility** | 3.1: Where there are trusted representatives involved (C), the programme brings credibility (MResO) which means stakeholders recognise the programme as having value (MResP). This leads to greater engagement in the work (O). |
| **4. Accessible spaces** | 4.1: Where there is no regular engagement with young people within the community (C), intentionally designing spaces for young people, with appropriate resource, and meeting their practical needs (MResO) can make it easy for young people to attend (MResP1) and helps young people feel seen and valued (MResP2). This leads to increased motivation to attend co-design sessions (O). |
| | 4.2: When a group of like-minded young people come together (C), facilitators intentionally using a range of techniques intended to promote accessibility and inclusion (MResO) makes the group feel safe (MResP1) and promotes openness and honesty (MResP2). This leads to active participation from young people (O1) and trust amongst the group/group cohesion (O2). |
| | 4.3: When young people have access to safe, accessible and engaging spaces (C), having group sessions to learn and talk together about shared local issues (MResO) leads to increased confidence and communication skills (MResP2). This improves their self-efficacy (O1) and leads to better insights to inform co-designed solutions (O2). |
| **5. Building a shared understanding/ mission** | 5.1: Where there is a constant pull towards individualistic ways of thinking (C), the programme team defining and keeping the focus on a locally situated social determinant (MResO) can help the co-design participants maintain shared understanding of the focus and boundaries of the work (MResP). This enables stakeholders to focus on locally specific areas of change that can improve adolescent mental health in the longer term (O). |
| | 5.2: Where there is safe and trusted space (C), bringing young people and community professionals together (MResO1) and having structured, facilitated conversations around a systemic issue (MResO2), gives young people and community professionals the opportunity to learn from one another (MResP). This helps to build connections and networks with one another (O1) and a shared understanding of lived experience and issues impacting young people's lives, locally (O2). |
| **6. Creating hope that change is possible** | 6.1: Where young people do not feel routinely heard (C), the programme team demonstrating that they are acting on young people's feedback and using this to shape the direction of the work (MResO) creates hope among young people that they can have agency in this space and contribute to change (MResP). This motivates them to keep attending and contributing to sessions (O). |
| | 6.2: When the programme includes the right people (C), a consistent champion can bring passion and commitment (MResO) which creates hope for change amongst community professionals (MResP). This motivates community professionals to continue supporting the programme (O). |

C = Context; MResO = Mechanism resource; MResP = Mechanism response; O = Outcome.

*"Oh, it's to make a difference. I've got little – I've got younger siblings and I've got nephews popping up here, there and everywhere. And I'd like to know that they might have a fighting chance of doing something they want to do, and not something they have to do"* (FG01_24, Young Person – Northern Devon)

PLOS Mental Health

*"There are some problems that we do face living in Newham that I thought if Kailo gives us a voice to express that and they made us feel heard and understood [...] I felt finally I have a voice in my community, I can do something good"* (FG04_24, Young Person – Newham)

Young people also talked about wanting to be part of the *Deeper Discovery and Co-design* sessions because it was an opportunity to 'get out the house' (FG01) or 'meet new people' (FG01). This was particularly the case in Northern Devon, where limited availability of youth groups and extracurricular activities made opportunities to meet and socialise with other young people rare.

*"[the reasons for showing up and taking part] well there was a few reasons. One because well, it's something to do. And two, because there was new people. Meeting new people, getting to try and make North Devon and the surrounding areas a better, more available place for people"* (FG01_24, Young Person – Northern Devon)

For some young people in Northern Devon, being paid to attend the sessions was also important for initial engagement. This was particularly relevant given there were few paid employment opportunities outside the tourist season. However, young people emphasised that payment was not the primary reason they continued attending.

*"It just helps me out with my day-to-day life. And now I've got a job [...] even if we – if it was to carry on, I wouldn't just come for the money"* (FG01_24, Young Person – Northern Devon)

**Community professional level.** Community professionals described wanting to be part of Kailo because it aligned with their personal, professional, or organisational priorities. For some, participation reflected personal ambitions to explore opportunities outside their formal roles. For others, it was driven by professional goals to empower young people or organisational aims to integrate Kailo-related insights into their work. These different types of alignment fostered initial and continued engagement, as well as active promotion of Kailo among other stakeholders.

*"I try and approach my work where I do a lot of things that aren't necessarily in the remit of my role itself, but opportunities to push myself in different directions [...] the idea that I could speak candidly with people about how they want services to look would be really beneficial (a) for myself, and (b) for my service in general, from anything I can feed back about it"* (S7_24, Community Professional – Northern Devon)

Kailo's methodological focus on youth voice and meaningful engagement with young people was one of the main reasons community professionals wanted to get involved in the programme, aligning with their professional priorities around youth engagement.

*"When you have young people supporting the work to improve the lives of other young people, I believe that there's more – the success rates are higher. Because they understand what the other people – what the community – people their age are looking for [...]. The fact that it was youth empowerment and having – engaging young people made a huge difference to our decision [to get involved]"* (S18_24, Community Professional - Newham)

This was particularly important in Northern Devon where there were few existing opportunities for meaningful youth engagement and a strong community need to hear and capture young people's voices. This was a gap Kailo identified during *Early Discovery*, leading to a greater emphasis on youth voice as Kailo's unique selling point, and a gap that aligned with young people's desire for more social activities within the community.

*"[It was] the synergy for me, it was perfect timing. A perfect storm kind of moment, it was some of the discussions we had been having in that group. [Name] had invited [Kailo facilitator] to present, and it was like 'oh great, now we have a*

*vehicle to do that'. So I was already very passionate about it. [...] [it] was the bit that we hadn't done is any engagement with young people to find out why"* (S6_24, Community Professional – Northern Devon)

This alignment with wider community need motivated community professionals in Northern Devon to continue their involvement, even after the Kailo sessions concluded. For example, two community professionals continued working with several of the young people to pursue Kailo inspired initiatives, while the wider community sought Kailo's support to establish a youth voice group to foster sustained engagement with young people.

While young people's voices were valued in Newham, several pre-existing initiatives already promoted youth voice and engagement. This meant Kailo's emphasis on this was less impactful. While Kailo may have filled other areas of need in the community (e.g., in terms of mission to address social determinants or use of evidence to inform practice), Newham did not have time to develop an understanding of these gaps (see **Time**) and emphasise how it might align with these. This resulted in reduced appetite for the work.

*"I think Newham has a mechanism, a council-wide mechanism already for hearing from young people within Newham. It had that prior to Kailo coming along. And, I... I think there is a particular structure to that which I don't think they probably feel needs much adaptation, or certainly not resourced to deliver it differently. [...] It either needs to be filling a gap that doesn't exist but where there is... motivation and resource to fill it"* (K4_24, Kailo Delivery – Newham)

**System leader level.** System leaders also talked about wanting to be involved in Kailo because of the clear alignment with community or organisational need, particularly around youth voice (i.e., Kailo methods) in the context of Northern Devon.

*"So we were really excited to be approached to put ourselves forward, to be a partner within the org- a delivery partner within the organisation because of the ethos of engagement of the young people and the co-design of young people within the process. That is rare. [...] But it's extremely rare that when designing a service or a system that anybody co-designs in a meaningful way"* (S15_24, Service Manager – Northern Devon)

However, some system leaders struggled to see clear alignment with their own priorities. At times, this was because points of alignment were not identified and clearly communicated by the Kailo delivery team. This led to limited or unsustained engagement.

*"I think that a few people, probably who maybe work more operationally, I would say some people have found it a bit difficult to grasp actually what the relevance is to their operational work to do with service delivery [i.e., people in charge of reporting outcomes]. So I think that's also been a bit of a – that's been a challenge. [...] The direct application of it [Kailo] to the work they are doing now has been a bit difficult for some people to grasp"* (K4_23, Kailo Delivery – Newham)

*"At the end of the session, someone asked a broad question around something completely unrelated to the work Kailo was doing (was clearly something they wanted to know from the research). This is part of the issue that everyone already comes with quite specific agendas/ideas about what they need to focus on based on other pressures"* (Observation, Big Circle Meeting 2 – Northern Devon)

This challenge was exacerbated by capacity and resource constraints. For example, one system leader in Northern Devon said that they were unable to take part in an interview for the Kailo evaluation as Kailo 'did not align' with their current organisational priorities. In Newham, while several system leaders were aware of the work, they were not closely involved nor consistently engaged.

### 1.2. Alignment with community professionals' daily work

**Community professional level.** Some community professionals specifically got involved in Kailo as they recognised an alignment between the Kailo activities and their daily, professional work. These community professionals talked about benefiting from being involved in the Kailo co-design process as it helped them: (1) learn about effective facilitation techniques; and (2) gain insights from young people. Some community professionals had taken this learning back to their workplace.

*"I learnt more culturally as to what to expect for young people as a group, the way people were talking and conversing, the things that they were conscious of or found important. [...] it was just interesting to see what are seen as priorities by younger people culturally at the moment, and that will help inform the things that I consider when I am working with younger people with additional need or not around that support. And, I guess it can help inform my approach better"* (S7_24, Community Professional – Northern Devon)

## 2. Time

### 2.1. Time to demonstrate commitment and build trusting relationships

**Community professional level.** The long-term nature of Kailo meant that the Kailo delivery team had time to embed themselves within the community, listening and responding to the views of community stakeholders. This investment of time, and intentional strategies employed within it, were critical to demonstrating commitment and building trusting relationships.

*"I think there's something really important and a bit cliched, but time allows you to move at the speed of trust [...]. You know, a lot of the underpinnings of the work have been very relational, understanding different perspectives, views, needs, where organisations are at, what they've learnt, giving time and space to surface those. And that just does take time to hear that and to like gather those multiple like views and perspectives. But there's also, time also buys you the ability to build trust by demonstrating that you're listening and responding to what people are sharing. It's like time on its own isn't the thing, it's the concrete demonstration of like trust and respect and also the kind of valuing of insights that have been developed and shared"* (K2_24, Kailo Consortium – Newham/Northern Devon)

In Northern Devon, this time and consistent presence resulted in the Kailo delivery team developing a deeper understanding of community need, identifying community stakeholders aligned with Kailo, and demonstrating a clear commitment to the local context.

*"The group talked more about potential delivery partners [for the Deeper Discovery and Co-design small circle sessions]. A lot of the suggestions were people [Kailo facilitator] had already thought of and had had some contact with. [Kailo facilitator] demonstrated a really good knowledge of the local area and some of the initiatives already happening which shows community partners her investment in Northern Devon. I think this really adds to the legitimacy of Kailo within the community. Having someone leading the work that is able to develop these connections within the local setting is essential."* (Observation, Community Partner Meeting 20/02/24 – Northern Devon)

In Newham, changes in the Kailo team disrupted this consistency, hindering relationship development and the identification of community needs that Kailo could address (see **Alignment**).

*"When I say this, I'm very aware that for example one of the things that are important for creating and building relationships is that it is the same person. Newham didn't have that. Newham had initially some people who are not there all*

*the time […]. Then they changed teams. There was a big hiatus of connection and relationship-building. People were not there. That thing that we were saying about time and what happens within that time, it wasn't happening" (K1_24, Kailo Consortium – Newham/Northern Devon)*

**System leader level.** Consistent presence was also challenging at the system leader level in both sites, as accessing system leader spaces was difficult and leaders often had limited time for meetings or flexibility in meeting agendas to discuss Kailo. This created barriers to building relationships with system leaders and engaging them in Kailo's work.

*"We're not in the right spaces yet. We had to build those relationships to be in those spaces, etc. However, in terms of being able to access the actual local community and the organisations that are there, that was something that we could do because it was easier to access them and be able to occupy spaces where they were. [...] It's harder to do that in a systems leader space, where you step into that space and occupy it." (K1_24, Kailo Consortium – Newham/Northern Devon)*

In some cases, only one or two relationships with system leaders were established. Staff turnover in Newham meant these relationships were not sustained, and additional time and presence would have been required to re-build connections with new system leaders.

## 2.2. Time and openness to learn from community and adapt processes

Having time and being open to learning and adapting Kailo activities to meet community stakeholder needs created opportunities for community stakeholders' voices to be heard.

**Young person level.** At the young person level, flexibility during co-design sessions in both sites was essential. It allowed Kailo to adapt session content and facilitate meaningful conversations with young people.

*"Being able to be flexible within a session and adapt to what's happening, or energy levels, or realising you need to build in another break, or stuff like that, that is definitely important [...] the flexibility, I think, was necessary if you were going to really do true co-design" (K15_24, Kailo Delivery - Newham)*

*"I think it opened up a space for lots of conversations that might not have happened if we were super strict with it. We obviously have a time schedule for sessions themselves and also a timeline for the points we wanted to be at within the process, but it was never super rigid. […] Session plans were also never planned out super ahead of time which I think has allowed us to take what has been done in previous sessions and working off that while incorporating it into the initial plan" (K17_24, Kailo Delivery – Northern Devon)*

In Northern Devon, this flexibility extended to adding additional co-design sessions to surface young people's voices, while in Newham, it included taking a break in co-design sessions for Ramadan. Understanding, responding, and adapting to young people's needs meant young people's voices were heard and ultimately that young people felt valued and willing to take part, generating richer local insights into the opportunity areas under consideration.

*"Even though they are technically the people who run it, they don't interrupt. If you need to get something across, they'll just sit and they'll write it down, or they'll put the recorder in front of you and they'll just head out and then... and you can see, when you come back to the next session, "Oh, I said that". And it's all there. They haven't missed bits" (FG01_24, Young Person – Northern Devon)*

**Community professional level.** Flexibility was also valued at the community professional level. In Northern Devon, one community professional talked about how their thoughts were valued and acted upon to create a better environment for young people to share their ideas.

*"[The Kailo team] were really open, like as if this was a dynamic process. [...] In similar situations I can imagine it being quite hard to positively challenge someone's approach to delivery, for example. Whereas it seemed quite hard wired into their approach to go, "Do you think this is working? How can we do things differently?" In terms of actual delivery with young people. So that was a really good working relationship" (S9_24, Community Professional – Northern Devon)*

However, there were also instances where this did not happen. For example, in the early stages of Kailo in Newham (i.e., while recruiting community professionals to support the co-design sessions), time pressure to progress the co-design sessions presented a more rigid context which meant they were not able to provide the same level of flexibility and adaptation.

*"At the start, at least, that [i.e., being flexible] was not the case. We needed to be a bit more rigid in terms of what it is that we need to accomplish by, and when. And that was primarily driven by the Kailo – the wider Kailo programme" (K24_24, Kailo Delivery - Newham)*

This led to problems recruiting community professionals to co-facilitate the sessions, as the application process required a quick turnaround and lacked flexibility to offer training.

*"We had one organisation that said, "Look, we'd really want to be involved, do you provide training for our staff to carry out some of the tasks that you want us to do?" But that wasn't within the scope of what we could offer, so that was a shame. And so that was one barrier, and two was, as I said, some small organisations have limited time for a quick turnaround. So I think that also meant that some organisations weren't able to submit their application." (K24_24, Kailo Delivery - Newham)*

### 3. Credibility

#### 3.1. Involving trusted representatives in the programme brings credibility

Young people and community professionals emphasised the importance of involving trusted local community members, organisations, or academics in Kailo (i.e., someone they thought highly of or had a good relationship with). This enhanced the programme's perceived value and encouraged greater engagement.

**Young person level.** All of the young people who participated in the *Deeper Discovery and Co-design* sessions were invited by adults with whom they already had established relationships. As well as creating a sense of the programme having value and being a worthwhile use of time, this also contributed to young people feeling comfortable and safe to attend.

*"Well, it was [Youth Worker] [...] I used to talk to her all the time, like every day. And she knows everything about me, so I think I felt a lot more comfortable with [Youth Worker] here" (FG01, Young Person – Northern Devon)*

Some young people also noted that they attended co-design sessions because their friends were involved, further reinforcing their sense of comfort and trust.

**Community professional level.** Community professionals often become involved in Kailo after being invited by colleagues or other professionals they knew or with whom they had strong working relationships.

*"It was an application that was led by [Community Partner]. And [Community Partner] reached out to us and [Community Organisation] to come together. We've been working together for years within the community [...]. We wouldn't have heard about it without [Community Partner]. Yeah, so our involvement is because of [Community Partner]." (S16/17, Community Professional – Newham)*

However, this was not important for all community professionals; others were invited by the external Kailo delivery team and only realised local colleagues were involved when they attended events. This was seen as a positive, confirming that they were a good fit with the work of the programme.

Some professionals also felt that the scope of different organisations/community members involved lent weight to the programme and created a sense of value and potential momentum.

*"I was pointed in the direction by my Director of Youth Work. [...] It felt it had weight because there was buy-in from other trusted members of the community. [...] It kind of felt like there's more likely for something like that to happen because there was, you know, it wasn't just us in a little room sort of thing." (S9, Community Professional – Northern Devon)*

For many community professionals, the extensive involvement of young people gave the project additional credibility. The opportunity to work with young people on co-designing strategies to address local issues was a significant motivator for many community professionals.

*"I thought this is good because we are going to get the young people to talk about their experiences and I think for our community, that doesn't always happen and often they are not heard. So, I was quite excited that they would have an opportunity to talk about their experiences so people can really understand what the gaps are" (S10, Community Professional – Northern Devon)*

Some community professionals also trusted the external Kailo consortium (consisting of universities and academic partners) to deliver a high-quality programme of work because of their reputation. Professionals were excited about the prospect of being involved with a research project and felt that the academic evidence underpinning the approach added value.

*"It is an outside team of people coming in so their lenses are fresh, not already kind of skewed with what might be wrong. So that was coming from a different angle, and I think the psychological side coming from UCL it was big enough for me to go OK if UCL are behind this, then this is something" (S10, Community Professional – Northern Devon)*

## 4. Accessible spaces

### 4.1. Practical ways to make young people feel valued

**Young person level.** The Kailo delivery team allocated substantial resources (financial and staffing) to designing and facilitating spaces that were suitable and accessible for young people. There was a strong focus on removing challenges around accessibility and this led to young people feeling their time and input was valued in a way that was not common. Examples of this included paying young people a living wage for their time, providing food, and providing and coordinating transport to the sessions (particularly in rural Northern Devon).

*"As someone who isn't working and is living off of benefits, I extremely appreciate the food and the money and the transport, so I am super grateful." (FG03, Young Person – Northern Devon)*

When asked about being paid to participate, young people expressed mixed views. While some emphasised the importance of payment others said they would attend sessions without it. Regardless, payment was universally seen as a positive incentive.

*"It is something that I wanted to do, something that I had passion about, helping communities, because I am also learning something out of it. I am taking something for myself, I am developing skills, so for me that is a bigger pay off than money." (YP16, Young Person – Newham)*

Young people also appreciated the flexibility of session timings to accommodate their schedules and use of accessible and safe spaces. For example, sessions were held in spaces other than schools, and venues were chosen to ensure good physical accessibility.

*"For me personally, I am disabled, I have mental health issues, I feel completely accepted. They have made me feel really welcome. I am able literally to get into the space and honestly it's amazing. I love it because I have gone to places where I physically actually can't get into the space and I have had to turn away and go "Look, I can't do this, I can't walk up that flight of stairs" or whatever." (FG03, Young Person – Northern Devon)*

### 4.2. Facilitators creating an accessible and inclusive environment

**Young person level.** Young people described the importance of facilitators creating an accessible and inclusive environment, where people could participate in ways that felt comfortable for them. The Kailo facilitators employed various techniques to ensure inclusivity, such as developing ground rules in the first session and offering multiple ways to participate (e.g., using white boards or quiet spaces) to accommodate neurodiverse young people.

*"[Kailo facilitator] tried to create an inclusive environment (knowing in advance that many of the YP had been referred from [Community Organisation] and had additional needs). This involved explaining that YP could participate (or not participate) in any way they wanted to. Whiteboards and note boards were placed around the room for YP to use if they didn't want to talk, there was no pressure to share anything they weren't comfortable with sharing, and fidget toys and quiet spaces were also available to use. Several YP ended up using whiteboards, fidget toys, and quiet spaces throughout." (Observation, Small Circle Session 1 – Northern Devon)*

Facilitators consistently reiterated participation was voluntary and that there would be no judgement for choosing not to engage from facilitators or other young people. For example, in Northern Devon they employed a sticker system (green, amber, yellow) for participants to indicate their comfort level with answering direct questions.

*"When they ask questions, I could have said whatever and nobody would have judged. Like I think I mentioned, in one of the sessions I mentioned having a mental asylum in place, and yeah, nobody really judged me too much for that". (YP17, Young Person – Newham)*

Both young people and Kailo facilitators noted that known adults in the room played an essential role in creating this safe and non-judgemental environment. Community professionals involved in the co-design sessions were able to share advice with the facilitators on how to work with young people they already knew well, or were involved in facilitating sessions themselves. Community researchers (young people aged 16 – 25 who are living in the local area and employed by Kailo to support with delivery) also helped to bridge the gap between young people and adults in the co-design process, sharing ideas about how to engage young people and work with them during the sessions.

*"I think also community partners, so having a youth worker in that space, or having a community representative who already has an established relationships with the young people in those groups. That was another way for them, they could call and check in on them, make sure they feel good and supported and if there is anything they need to feed back to us they could do that at any point" (K1, Kailo Consortium – Newham/Northern Devon)*

Over the course of the co-design sessions, this sense of safety led to greater engagement from young people and willingness to contribute to discussions.

*"There was lots of positivity in the room and comments about feeling relaxed, safe and comfortable. It is worth noting that not all of the YP who joined the sessions at the start were present, so clearly some young people didn't/couldn't continue attending. It seems that [Kailo facilitators/community professionals] have created a space and group dynamic where most YP and community members feel safe, valued and able to contribute to discussions." (Observation, Small Circle Session 6 – Northern Devon)*

The young people, most of whom had never met before, also formed friendships and developed trusting relationships with each other.

### 4.3. Improved self-efficacy and confidence in young people

**Young person level.** Participating in the Kailo co-design process led to improved confidence and communication skills among the young people involved. Feeling safe in the group allowed them to engage in meaningful conversations with adults about local issues affecting youth. This contributed to young people developing higher self-esteem.

*"The young people there, I personally know many, were very shy and timid, and by the end of the project they were very open and able to express themselves better and communicate better. And they're not shy to speak to people anymore, they work much better in teams now than they did at the beginning of Kailo which is very positive to see." (S18, Community Professional – Newham)*

Young people feeling able to be open and confident when discussing local issues and developing design ideas led to better insights for the co-designs than if adults had been trying to solve problems without youth input.

"[When community partner suggested bike racks and bikes as a solution to transport issue] Two of the YP quickly said that there are some bike racks and more would be a terrible idea because a) your bike would get nicked, and b) you would get bullied if you cycled to school. One YP also said she had never learned how to ride a bike. [Kailo facilitator] asked how come, as surely it was something you all did in primary school? The YP replied that there were cycling proficiency sessions, but only for people who had their own bikes, and her family couldn't afford one.

*This felt like a good example of the importance of young people's voices in these discussions – the adults could easily have become carried away with the idea of bike racks, but the YP very quickly shut this idea down by explaining their experiences. It was also an example of YP's opinions being listened to and respected, as the adults acknowledged this and said 'fair enough' before moving on to other topics. The fact that the YP felt they could speak up (and that they were bothered to do so) I think shows a level of comfort in the group and belief that their voices will be heard and respected." (Observation, Small Circle Session 6 – Northern Devon)*

## 5. Building a shared understanding/mission

### 5.1. Maintaining boundaries of the work

**Young people and community professional level.** Everyone involved in the *Deeper Discovery and Co-design* phase of Kailo reflected that a focus on a locally situated social determinant is important for maintaining the boundaries of the work. A crucial step in this process was the prioritisation and definition of the "opportunity area" (specific locally relevant priority areas linked to a social determinant) to address. The success of this process varied across Kailo teams

and sites. In some cases, co-design sessions began with a very clear and well-defined topic, such as "diverse job opportunities" in Northern Devon and "violence and crime" in Newham (see Table 1). Those involved in these sessions had a clear idea of the problem they were working on and designing solutions for. This clarity also meant that the Kailo teams were able to recruit both young people and community professionals who were motivated to contribute to a specific issue.

*"The young people were asked to do a quick design icebreaker, creating a game or a curriculum on something they were interested in. Despite being told it didn't need to be Kailo-specific, all young people decided to focus their designs on addressing the topic of improving access to job opportunities. It seems as though they are all really clear on the focus of the work they have been doing in the sessions and are already thinking about potential solutions for young people, parents, and schools." (Observation, Small Circle Session 8 – Northern Devon)*

In contrast, broader opportunity areas less obviously linked to a social determinant such as "mental health awareness and literacy" in Northern Devon and "activities for wellbeing" in Newham (see Table 1) posed challenges. These topics made it difficult to maintain a focus and foster a sense of working towards a shared goal.

*"I guess the question always raised in my mind is whether the opportunity area was framed the right way in the first place..." (K4, Kailo delivery – Newham)*

*"Community partners reflected that the young people don't know where they are headed with the co-design sessions (and neither do the community professionals). Some felt it would be useful to get some clarity about what they are working towards as the focus of the opportunity area has shifted" (Observation, Midway Reflection Session – Northern Devon)*

Close and skilled facilitation of the co-design sessions was also key to keeping discussions on track. Facilitators needed to work hard and consistently across the session redirect conversations and ensure sessions remained focused.

*"I feel as though who facilitates each group discussion is key for the success of the session. Where groups are facilitated by a Kailo team member who knows what they are doing, they can keep the group on task and help generate relevant discussion (as can go off track sometimes!). I think some members of the Kailo team have more experience doing this and are closer to Kailo aims than others. Community partner support can also be really helpful, as long as they are on the same page in terms of what they are trying to achieve within the session." (Observation, Small Circle Session 4 – Northern Devon)*

Some community professionals commented that it was also helpful to have an external team outside of the local service landscape leading the work. This neutrality helped the Kailo delivery team maintain a focus on prevention and social determinants, rather than being influenced by organisational priorities.

*"It also offers real opportunities because there are no previous affiliations... or reputations that are at play. It's an entirely neutral team that are coming in. I saw that as a real positive." (S12, Community Professional – Northern Devon)*

Underpinning all of this was the importance of Kailo facilitators having a clear understanding of social determinants and potential solutions for addressing structural issues in local communities. This was not always the case in the first phase of Kailo, which led to some confusion amongst the team and challenges around communicating and facilitating the work.

*"It isn't something that we've talked about in the recent, since Christmas... And I think that is one of the things that I'm unsure that we've achieved... whether we got them to focus on the right factors or not... Yeah, that's something I'm unsure of, if the [Small Circle] group they're truly addressing a social determinant of health." (K15, Kailo Delivery – Newham)*

### 5.2. Structured facilitated conversations give opportunity to learn from one another

**Young people and community professional level.** In the context of a safe and trusted space, young people and adults benefited from the opportunity to come together to have structured conversations around the social determinant under consideration. For young people, these conversations provided an opportunity to learn about others' experiences and perspectives, particularly around mental health.

*"I guess I was made aware because I'm older, of how different people deal with it [mental health]. Obviously all my mates are a very similar age and we've all got the same way to deal with when we're having a bit of a shit day or whatever. But [young person] would deal with her seasonal depression in a different way than I would. So it was – yeah, it opened my eyes of how different people would do it and how different people cope." (FG01_24, Young Person – Northern Devon)*

For community professionals, these conversations deepened their understanding of young people's experiences in the local area and the extent of the issue.

*"[The focus on social determinants] I think it's highlighted some themes. It's enabled us to get some clarity about what is really going on for people right now and their perceptions of those things, those determinants." (S12_24, Community Professional – Northern Devon)*

For everyone involved, these discussions fostered networking and connection, allowing participants to build relationships with like-minded people.

*"Young person 1: I love spending time with these idiots, said lovingly, I mean it lovingly. I love these idiots, they are my idiots, I am the group mother [...]. Young person 2: I feel accepted. [...]* **Facilitator: Accepted? Can you say why? What makes it-**. *Young person 3: Just non-judged. Young person 2: Yeah, I don't feel like I have to pretend. Young person 1: And being around like-minded people." (FG03_24, Young People – Northern Devon)*

*"The professionals have clearly got to know each other well through the small circle sessions – attending each other's events and sharing advice with one another. I wonder how well they knew each other beforehand. My understanding is they might have known of one another but not interacted much. Seems like one of the key things that the small circle sessions allow for – that bringing together of professionals, giving them an opportunity to network and learn from one another, while working towards a shared goal. Don't get these opportunities otherwise?!" (Observation, Small Circle Session 15 – Northern Devon)*

The Kailo delivery team used various methods to facilitate these conversations effectively, including individual and group activities, creative methods (e.g., playing games, art and storytelling), and group model building sessions. The latter involved tasking participants with creating a systems map, thinking about how different aspects of the wider system contribute to the social determinant of interest. These sessions were facilitated by systems thinking experts and helped young people and community professionals take a step back and think together about how the wider environment linked to specific issues.

*"[…] I had to stop and discuss with some other person, why do you think this is more important than this one? What do they think? Getting their input and their point of view and the personal things that happen to them. It also made me see a different way, so that one was one of my favourite sessions because that one it really made me think"* (YP16_24, Young Person - Newham)

*"Community partners were enthusiastic about the systems sessions, saying that it was great for young people to be encouraged to see the world in different ways. Adults were impressed with how quickly the young people grasped the key ideas (although some of the language was too technical) and a few young people have reported using this new way of thinking outside of the session"* (Observation, Community Partner Meeting – Northern Devon)

*"[talking about what they learnt from systems thinking sessions] I guess it all kind of links everything. And if you improve one thing it is easier to improve other things [...]. I think as well I think it also highlights that – it is very complicated and the system shows that you can't just expect young people to do this when in fact it is not just down to young people, it is down to other companies and schools. And as I say the government – we all need to work together [...]. It kind of shows this is what we need to help young people and for the future"* (FG02_24, Young Person – Northern Devon)

While these structured conversations were valuable, their success depended on effective facilitation. In some instances, a lack of capacity to facilitate sessions or clarity about the focus of the work from the Kailo delivery team resulted in confusion about the task at hand (see **CMOC 5.1**). In particular, large groups were difficult to facilitate effectively, leading to disengagement and an inability to stay focused on the systemic issue at hand.

*"[Kailo delivery] said that they had a bigger group expected in Barnstaple (nice that they keep wanting to come back!) and, as a team, they had been discussing how it can be a challenge in terms of getting a collective group dynamic (with group breaking off into three). It is also challenging in terms of ownership; groups are resistant to work with one another which means ideas are isolated within individual groups which makes it hard to think collectively about how these ideas come together. In turn, this leads to confusion about direction." (Observation, Community Partner Mid-Way Brief – Northern Devon)*

Furthermore, issues such as consistency of the group members affected whether young people and adults felt comfortable to have these group conversations. For example, in Newham one of the co-design sessions had very inconsistent/fluctuating membership, with new young people joining the group regularly. This resulted in a space where participants did not feel safe to discuss issues and learn from one another.

*"Because of the waxing waning membership, I don't know that we've ever really got group consensus around some of these things [co-design ideas]. So it's been really hard to get consistency and understand that this is something that actually is really important to a representative number of people, or it's just something that someone has talked very passionately about and people have gone "Yeah, that will do""* (K18_24, Kailo Delivery – Newham)

## 6. Creating hope that change is possible

### 6.1. Acting on feedback from young people creates hope

**Young person level.** Young people felt that their opinions and views were acted upon and fed into sessions, creating visible signs of progress, trust in the process, and hope that it could lead to change.

*"...the fact that you are progressing on the same project every week and you are working on what you have already done and trying to move forward to the next step. I think that was motivating in knowing that you are improving in something every week, every two weeks". (FG04, Young Person – Newham)*

The Kailo delivery team would often remind the young people that their work was useful by consistently thanking them for their thoughts and reflections and referring to them as:

*"Key collaborators, equals and decision makers in the (co-design) process" (Observation, Small circle 8 – Newham).*

Involving young people from the beginning and demonstrating that young people were shaping the direction of the work motivated some participants to keep attending the sessions and contributing their ideas. This created hope that the designed strategies could lead to change.

*"Yeah, I completely agree with that [wanting to see change happen]. I think the idea of wanting to make a change and then going along with it every week and then thinking about it, thinking you could talk about this or you could add this to it, I think every week it was building up to thinking that you are excited for what the big outcome would be." (FG04_24, Young Person - Newham)*

*"**Interviewer: What is it about Kailo that makes you think something might change?** Young person 1: That you spoke to young people about it, instead of adults and teachers and people like that. Young person 2: You built a relationship with the young people first, then you got the professionals in, instead of doing it the other way round. So you actually got the voice of what young people want and then you got a professional to talk to them. I think that is the proper way to do it." (FG01, Young People – Northern Devon)*

However, others were unsure about the potential impact of their work because of the scale of some of the potential solutions and the fact that the Kailo team had not communicated how these barriers would be overcome.

*"Interviewer: do you believe that the work that you have done will lead to change in your local area? Young person 1: I hope so [laughs]. I wouldn't say it is one hundred percent confirmed because you never know, but I think it is a good starting point [...]. Young person 2: I feel it will, but I am just wondering how they will do it I guess, because funding is quite a big thing and you will need to probably get government help and stuff. So I am just wondering and say in schools how you would enforce it with the support for young people" (FG02_24, Young People – Northern Devon)*

In one instance, young people reported feeling unheard when a community professional became defensive about their organisation. This experience undermined their sense of hope for change.

### 6.2. A passionate champion creates hope among community professionals

**Community professional level.** Some community professionals felt hopeful that the Kailo work would be taken forward and create change in the local area. Community professionals often linked their hope for change with the passion and dedication demonstrated by the Kailo delivery team.

*"I am very optimistic that the team that you have got, that are involved or connected with, are passionate enough, inspired enough to create something out of it" (S6_24, Community Professional – Northern Devon)*

Having a consistent champion (whether this be the Kailo delivery team or community stakeholders) to demonstrate this passion and commitment was important for motivation and continued engagement in Northern Devon. However, in Newham, a change in personnel during the first year of the programme meant that there was not the same consistency.

*"We had team changes in [Kailo facilitator 1] coming off the project and... [Kailo facilitator 2] coming on and off at different points. And I think that, I mean, it's well understood, but really good deep community work requires consistent*

*relationship holding, and yeah, even by a month into my time, I could see that that was really a big challenge and would have required a lot of energy on our side to pick up and rebuild those relationships" (K5_23, Kailo Delivery – Newham)*

Having a range of relevant people involved in the project also contributed to community professionals' hope that structural change could be achieved.

*"I think the processes informed and educated practitioners and professionals in a really – across a really wide space. So that will influence their decision making. It will influence what they want to do and where they think they can go. And to look at things differently. So that's a really good thing". (S9_24, Community Professionals – Northern Devon)*

However, for some, this did not go beyond hope as Kailo had not clearly communicated the next steps or evidenced how they were going to take the work forward.

*"It felt like sometimes it might have just been a great method for strategising and building your own knowledge, but how much is going to be sustained later on?" (S20_24, Community Professional – Newham)*

Community professionals also highlighted barriers in achieving systemic change such as funding and capacity. Despite passion and enthusiasm from the Kailo delivery team, these practical barriers to implementing strategies left some community professionals feeling less optimistic about the possibility of change.

## Discussion

Our realist-informed developmental evaluation examined how Kailo – a place-based approach to addressing adolescent mental health and wellbeing – functions as an initiative, for whom, and under what circumstances (**RQ1**). We also explored the contextual factors and conditions necessary for place-based systems change (**RQ2**) and the outcomes prioritised through this approach (**RQ3**). To address these questions, we developed 12 IPTs which we organised into six themes: (1) Alignment; (2) Time; (3) Credibility; (4) Accessible Spaces; (5) Building a Shared Understanding/Mission; and (6) Creating Hope that Change is Possible. The findings have informed Kailo's ongoing development and offer valuable insights for the broader field in terms of how complex place-based approaches operate. Below, we discuss our findings in relation to our three research questions before proposing a middle-range theory for Kailo.

1)  How does Kailo function as an initiative? Why and for whom?

Our evaluation identified several mechanisms that are critical to Kailo's functioning (**RQ1**). These include demonstrating **alignment** (CMOCs 1.1 and 1.2), **flexibility** and building **trust**ed relationships with the community (CMOCs 2.1 and 2.2), **credibility** (CMOC 3.1), **enabling participation** through the provision of safe and accessible spaces (CMOCs 4.1, 4.2, and 4.3), **fostering a shared focus** on locally situated social determinants (CMOCs 5.1 and 5.2), and **acting on feedback** and **championing** the work to create hope for change (CMOCs 6.1 and 6.2). When these mechanisms were activated, community professionals and young people were motivated to engage with Kailo throughout the programme, able to build a shared understanding of the focus of the work and empowered to contribute to system-change efforts. These insights add to the limited evidence-base on how and why place-based system-change approaches operate [24]. Specifically, they illustrate how key components of place-based approaches, such as building partnerships and community engagement [14], might be achieved. Consequently, these findings offer actionable guidance for designing similar place-based initiatives in the future.

While these mechanisms largely align with how the Kailo consortium hypothesised Kailo might function [27], a significant gap emerged regarding the utilisation of data as a driver of systemic change. Data use (e.g., local data to inform needs identification and prioritisation and wider academic data to inform the design/adoption of strategies) is frequently

cited as a critical component of system-change efforts [14,45] and is a core feature of other place-based approaches, such as Communities That Care [18,20]. We initially anticipated that synthesising administrative and epidemiological data with qualitative insights from community engagements would play a central role in informing co-designed solutions [27]. However, challenges in accessing meaningful and locally applicable data and integrating these data into Kailo's phases hindered this ambition. Moving forward, it is essential that we consider how Kailo might better balance the use of data and research evidence with a bottom-up approach that fosters local innovation to ensure efforts effectively inform and sustain systemic change.

2) How is Kailo received in a local context and what conditions are necessary for place-based systems change to be achieved through Kailo?.

We also identified several contextual factors that influenced whether mechanisms (identified as necessary to Kailo's success) were fired in Newham and Northern Devon (**RQ2**). Three critical factors were time, consistency, and being open to learn and adapt. These contextual factors enabled the Kailo team to actively listen, adapt, and respond to the needs of the community and, in turn, develop an understanding of the local context, engage with the 'right' people, and respond to gaps in need (CMOC 2.1 and 2.2). Another important factor was salience of the programme to local community stakeholders, which was essential to ensure the community were motivated to engage with the work long-term (CMOC 1.1). This could not be achieved without a clear understanding of community need, with notable 'ripple effects' [46] across these CMOCs.

Similar factors – such as having sufficient time to engage with the community, adapt to community need, and build relationships – have been identified as essential in other place-based approaches (e.g., [9,10,47,48]). Time and resource to engage with the community can be conceptualised as critical "readiness" criteria for future Kailo sites. However, we also identified conditions that could hinder Kailo's ability to create systemic change, namely the "pull towards individualistic ways of thinking" (CMOC 5.1). This issue stems from a societal context which focuses on treatment of individual needs and behaviours rather than a social model of mental health and wellbeing [49]. This challenge is not unique to Kailo; similar issues have been reported in other structural change initiatives, where solutions often shift toward focusing on the individual level (e.g., [50]). Within Kailo, it was essential to foster a clear focus on a locally situated determinants (recognising the link to national drivers of poor mental health and wellbeing) to ensure co-design session participants maintained a shared understanding of the focus and boundaries of the work. Nonetheless, sustaining this focus proved difficult.

Further efforts are needed to ensure structural solutions remain the central focus in place-based approaches like Kailo. This might involve identifying and engaging system leaders earlier in the process, fostering a youth-centred approach which draws more substantially on other community stakeholders to help inform the direction of the work, or utilising a co-design methodology more focused on the development of structural solutions. Moreover, additional consideration is needed on how the implementation of place-based system change approaches might be hindered by national drivers that shape local decision-making, funding, and resource allocation, ultimately contributing to system-level challenges. These national policy and regulation drivers transcend geographical locations and need to be better considered and addressed within place-based approaches, including Kailo, to ensure place-based systems change can be achieved.

3) What outcomes are prioritised in a place-based system through Kailo?.

Within the first two phases of Kailo, the prioritised outcomes (**RQ3**) were primarily process-orientated or intermediate, related to Kailo's own engagement and functioning and highlighting benefits for individuals directly involved. Key outcomes included engaging the right stakeholders (CMOCs 2.1), increased motivation to participate and sustained engagement in the work (CMOCs 1.1, 2.2, 3.1, 4.1, 6.1, and 6.2), enhanced self-efficacy and confidence among the young people involved (CMOCs 4.3), increased trust and collaboration between community stakeholders and young people (CMOCs 2.1 and 4.2), and a shared focus on social determinants (CMOCs 5.1 and 5.2). A better collective understanding of the local community and local issues impacting young people's lives was also evident (CMOCs 2.1, 2.2, 4.3, 5.1, and 5.2),

 

with community stakeholders championing Kailo and integrating their learning into their everyday practices (CMOCs 1.1 and 1.2).

These process and intermediate outcomes are consistent with findings from other place-based approaches [33]. For instance, the *Agenda Gap* [8] – a youth-centred mental health promotion and policy advocacy programme – highlighted similar outcomes including the empowerment of young people, sustained interest and engagement in the programme, and increased trust between young people and adults as important [8,51]. However, a persistent challenge in place-based approaches, including Kailo, is determining how these process or intermediate outcomes translate into systemic change or distal, population-level impacts [33]. While many place-based approaches emphasise the empowerment of young people or the capacity-building of adults working with youth [8,9,47,52–54], the pathways linking these efforts to long-term improvements in adolescent mental health outcomes remain uncertain [33]. Achieving these distal outcomes is likely to require time, structured support and guidance from the project team, and ongoing system-wide support, even after the project's conclusion. This involves ensuring that lessons learned, and processes initiated during the programme are embedded within the broader system to enable lasting change. Moving forward, further refinement of the Kailo framework will require a deeper understanding of how these process outcomes contribute to systemic change and subsequently drive long-term improvements in population-level adolescent mental health. Exploring these pathways will be a central aim of our upcoming impact evaluation, which seeks to build on the insights from this developmental evaluation.

### Further development of IPTs

The programme theories generated through our developmental evaluation help to explain how, why, and for whom Kailo is functioning as an initiative. However, it is important to acknowledge these theories are not fixed in time and are likely to evolve as Kailo engages with new sites, where differing community contexts and needs will shape its implementation. In particular, future work should consider how we can apply an equity lens to our CMOCs. Frameworks such as PROGRESS [55] highlight the importance of considering factors that contribute to health inequity (including place of residence, race/ethnicity/culture/language, occupation, gender/sex, religion, education, socioeconomic status, and social capital) within health interventions and programmes. Applying this lens could help us explore how power imbalances, systemic inequities, and social identities might further shape the operation of our CMOCs. This is particularly critical as Kailo extends into new sites where contextual differences may vary significantly. Incorporating an equity perspective will ensure the adaptability and inclusivity of Kailo, enabling it to address diverse community needs more effectively while fostering sustainable systemic change.

### Towards a middle-range theory for Kailo

Several middle-range theories [56] could help us to understand and conceptualise aspects of the IPTs developed in this developmental evaluation. For instance, our CMOCs on alignment (1.1 and 1.2) can be better understood through the **Theory of Systems Change** [45] which suggests that aligning with the strengths and needs of the local-population is crucial for establishing well-functioning systems that promote improvements in population health outcomes. Additionally, **Third Space Theory** [57] and **Empowerment Theory** [58,59] provide useful framing for our CMOCs on accessible spaces (4.1 – 4.3). **Third Space Theory** highlights how creating safe spaces in neutral settings can help foster discussion and amplify otherwise un-heard voices [57], while **Empowerment Theory** explains how empowerment operates at multiple levels to enhance an individuals' or organisations' ability to act and improve health outcomes [58,59]. However, no single middle-range theory fully captures the breadth of insights from our developmental evaluation. This may be due, in part, to Kailo's innovative integration of place-based working with systems change. To address this gap, we propose consolidating our findings in an overarching middle-range theory for Kailo:

*"By aligning evidence-based, place-based, social determinants of adolescent mental health and wellbeing with local system priorities, a participatory approach that empowers community stakeholders through shared missioning has the potential to shift the structures that contribute to, or perpetuate, mental health and wellbeing difficulties".*

This theory serves as a foundational hypothesis for how and why place-based, systems change initiatives like Kailo operate. It remains open to further testing, refinement, and adaptation as new evidence and insights emerge [56].

**Strengths, limitations and future directions**

Our study benefits from utilising multiple sources of data collected from two geographically distinct sites, Newham and Northern Devon, to inform the development of the Kailo framework and its programme theory. It also benefits from utilising two YPAGs to ensure the methods and results were accessible and applicable to young people. This approach provides insights that could be valuable for future place-based approaches. However, our study is not without its limitations. First, we collected data from a subset of community stakeholders and young people and were not able to collect data from those who chose not to engage in Kailo at different stages. Understanding why some young people stopped attending co-design sessions would have been particularly useful to inform the development of IPTs.

Second, despite the intention to follow Kailo throughout one whole lifecycle of the framework [35], delays in initiating the programme in both sites meant we were only able to focus on the first two phases of the framework: *Early Discovery* and *Deeper Discovery and Co-design*. These phases represent what had been delivered up to July 2024. Delays were mainly due to the challenges of research funding timelines and an external team coming in to deliver the framework in Northern Devon and Newham; turnover of community leaders and professionals meant that some of the initial relationships (established before the main application for funding) were lost and it took time to rebuild these connections. As a result, we were unable to gather data on many of the hypothesised systems outcomes or broader population-level adolescent mental health and wellbeing outcomes. Instead, the programme theory reflects foundational and intermediate process outcomes, which are hypothesised to precede such systemic change. Future work should investigate the pathways through which these foundational and intermediate process outcomes translate into population-level improvements. Understanding these pathways will be essential for refining the Kailo framework and enhancing its effectiveness in achieving systemic change and lasting health outcomes.

## Conclusions

This realist-informed developmental evaluation has provided critical real-time learning that has informed the further development of the Kailo framework. The evaluation has generated programme theories and a middle-range theory that contribute to our wider understanding of how complex, place-based systems change approaches might function. Our work also lays the foundation for a future impact evaluation, planned for 2025–2026, exploring whether and how the framework contributes to broader systems-change that could subsequently lead to population-level improvements in adolescent mental health and wellbeing.

## Supporting information

**S1 Text. Description of and rationale for realist approach.**
(DOCX)

**S2 Text. Key Kailo terms and stakeholders.**
(DOCX)

**S3 Text. Types of activities observed as part of the developmental evaluation.**
(DOCX)

**S4 Text. Round 1 interview schedule example – Kailo consortium.**
(DOCX)

**S5 Text. Round 2 interview schedule example – Kailo consortium.**
(DOCX)

**S6 Text. List of initial CMOCs.**
(DOCX)

**S1 Fig. Timeline of Kailo phases and evaluation activities.**
(TIF)

## Acknowledgments

Thank you to all those young people, community stakeholders, senior leadership, and Kailo consortium members who participated in our developmental evaluation. Special thanks also to the members of our Young Person's Advisory Groups (YPAGs) who helped shape and support the conduct of this evaluation. This includes Ash W-S, Beth Vingoe, Brigit Duhig, Callum S, Chris Sims, Ejiro Mowoe, Lereece Partrick, Mish, Nevaeh Matthews, Roisin, and Willow M-B. Finally, thank you to Bekkah Bernheim for providing comments on our manuscript and to the Kailo Expert Advisory Board for their guidance.

This research was conducted as part of the Kailo programme and we gratefully acknowledge the valuable input and discussions from the wider team that have informed this paper.

## Author contributions

**Conceptualization:** Tim Hobbs, Peter Fonagy, Steve Pilling, Vashti Berry.

**Data curation:** Kate Allen, Anna March, Bianca Alexandrescu, Siying Li, Laura Kennedy, Karuna Davis, Tamanna Malhotra.

**Formal analysis:** Kate Allen, Anna March, Bianca Alexandrescu, Julie Harris, Rachael Stemp, Laura Kennedy, Karuna Davis, Tamanna Malhotra, Ediane Santana de Lima, Tim Hobbs, Niran Rehill, Jenny Shand, Steve Pilling, Vashti Berry.

**Funding acquisition:** Tim Hobbs, Peter Fonagy, Steve Pilling, Vashti Berry.

**Investigation:** Kate Allen, Anna March, Bianca Alexandrescu, Julie Harris, Rachael Stemp, Siying Li, Laura Kennedy, Karuna Davis, Tamanna Malhotra, Steve Pilling, Vashti Berry.

**Methodology:** Kate Allen, Anna March, Julie Harris, Rachael Stemp, Laura Kennedy, Steve Pilling, Vashti Berry.

**Supervision:** Julie Harris, Jenny Shand, Steve Pilling, Vashti Berry.

**Validation:** Julie Harris, Ediane Santana de Lima, Tim Hobbs, Niran Rehill, Jenny Shand, Steve Pilling, Vashti Berry.

**Writing – original draft:** Kate Allen, Anna March.

**Writing – review & editing:** Kate Allen, Anna March, Bianca Alexandrescu, Julie Harris, Rachael Stemp, Siying Li, Laura Kennedy, Karuna Davis, Tamanna Malhotra, Ediane Santana de Lima, Tim Hobbs, Niran Rehill, Jenny Shand, Peter Fonagy, Steve Pilling, Vashti Berry.

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
