## [Decision Letter · Decision Letter 0]

PMEN-D-24-00539

Developing programme theory for a place-based, systems change approach to adolescent mental health: A developmental realist evaluation

PLOS Mental Health

Dear Dr. Berry,

Thank you for submitting your manuscript to PLOS Mental Health and I am very sorry for the delays. After careful consideration of the reviewer reports we would like to invite you to submit a revised version of the manuscript that addresses the points raised during the review process. Please ensure that all points are addressed - these can be found at the end of this email. 

We look forward to receiving your revised manuscript.

Kind regards,

Karli Montague-Cardoso

Executive Editor

PLOS Mental Health

Journal Requirements:

Additional Editor Comments (if provided):

Reviewers' comments:

Reviewer's Responses to Questions

**Comments to the Author**

1. Does this manuscript meet PLOS Mental Health’s publication criteria? Is the manuscript technically sound, and do the data support the conclusions? The manuscript must describe methodologically and ethically rigorous research with conclusions that are appropriately drawn based on the data presented.

Reviewer #1: Yes

Reviewer #2: Yes

2. Has the statistical analysis been performed appropriately and rigorously?

Reviewer #1: N/A

Reviewer #2: Yes

3. Have the authors made all data underlying the findings in their manuscript fully available (please refer to the Data Availability Statement at the start of the manuscript PDF file)?

Reviewer #1: Yes

Reviewer #2: No

4. Is the manuscript presented in an intelligible fashion and written in standard English?

Reviewer #1: Yes

Reviewer #2: Yes

5. Review Comments to the Author

Reviewer #1: Hello,

Thank you very much for the opportunity to review this paper. I get very excited reading about place-based work for adolescents! It's on a really interesting topic and what seems like an impactful program. The paper is well-written and provides a lot of good insights into how place-based initiatives working with local communities can ensure that they have meaningful, long-term and fruitful collaborations with these groups.

That being said, there are some minor amendments and additions which I think to improve the quality of the work. Generally, I think that there is too much assumed knowledge in the current manuscript, which prevents the reader from fully understanding the work.

It would be brilliant if the authors could add a diagram which demonstrated the process undertaken in the development evaluation, and how each step fit in within broader Kailo program stages and objectives. It was sometimes hard to follow what was happening and a diagram could help guide participants' understanding.

In the introduction, there needs to be more of an explanation of clear constructs. For example what are 'place-based approaches'? How do they differ from alternative models and what is the strength of having a place-based approach. The current explanation that they are different to individual standardised interventions and that they acknowledge social determinants, is quite vague. Real-world examples would be helpful. Similarly, what is a 'systems change approach', how do they differ from the common orthodoxy? This needs to be bedded down before you go into discussions of complex systems later in the paper. I think you need to do more to take readers along for the ride. Terms like these, unless explained well and integrated well, can start to feel like buzzwords without meaning. Similarly 'joined up working' (line 74).

In line 94 - isolated to individual-level impacts for whom? For those who participate? And opposed to what? benefits for the whole community??

Generally I think you need to give more air time to Kailo as a program. What the program includes, what it doesn't and what it actually aims to do. I got a bit confused because the results section often appeared as if it was evaluating a naturally occurring phenomenon, without out really making clear that Kailo was a pre-defined and carefully developed framework which was delivered in the community. The readers would benefit from a bit more detail about what kailo was trying to achieve, so that the outcomes presented in the evaluation can then be compared against these aspirations. It was also unclear where the role of Kailo begins and ends. Is Kailo just a means of convening existing community members and organisations to have productive discussions about what's best for their communities (i.e. a co-design facilitator)? Or is Kailo also involved in the implementation of these ideas? Does Kailo fund and/or facilitate the proposed community outcomes (e.g., 'Life Skills training for young people'). I appreciate this paper references other papers which describe the Kailo process, but I do think that a bit more detail is needed in this paper to improve its quality as a standalone piece.

Finally, I think more detail is needed about how the young people engaged in the sessions were recruited. Were adolescents with lived experience of mental health issues or poor wellbeing targeted for inclusion - this seems to be an important component of the work. Also, what proportion of those who were approached to participate actually participate? It was mentioned that peers often encouraged eachother to come, and some informants were recommended by the Kailo delivery team, but some young people dropped out and never returned. There is of course a real risk when this happens that you get homogenous 'groupthink' from similar peers, and lose out from those who may have benefitted most from the program or provided the most useful feedback based on their lived experience. This feels particularly important when in the next stage of the project you try and understand the distal, long-term population-level impacts, particularly to the local young people most at risk of poor mental health. I think you need to provide more of a rationale in the discussion as to why you did not try and track down those who dropped out of sessions to understand why this occurred and if they had felt comfortable sharing their perspectives during sessions.

I really like the table, framing of the Themes and CMOCS and the way each is labelled with Context, MResO, MResP and Outcome.

Reviewer #2: This is a study of the implementation of a community-engagement framework for developing place-based solutions for youth mental health and wellness (Kailo framework). The paper is very well-written and it is timely as interest in place-based approaches addressing youth mental health appear to be growing. I appreciated the use of realist evaluation which illustrated useful CMO statements that I believe will be useful not only for theory-building but practically for those working on similar projects. I have a few comments that I hope might be helpful to the authors.

The authors could say more about why a participatory and design-oriented approach is needed if CTC has already demonstrated strong outcomes. I looked at the originating article on Kailo to better understand the overall framework and see strong justification in the 2023 open letter that could be brought into the introduction in this piece.

Unless there is another compelling reason, I don’t believe the authors need to frame their study as a developmental evaluation as it is sufficient to note that the study is a realist evaluation of the first two phases of a planned four phased implementation project. In describing this as a developmental evaluation, the reader might expect to see more in the paper on how the results were used to improve the effort as it evolved.

The methods would benefit from more operational discussion of the Kailo implementation infrastructure: Who is funder, who are the implementation agents, who monitors implementation alignment with the Kailo framework, and are any funds allocated to implementing plans developed in Phase two.

In the discussion, the authors note that there were delays in starting the project. I believe the reader would benefit from knowing whether those were true delays (eg funding or a project lead was not able to begin) or simply the natural/normal amount of time it will likely take other sites to adopt this type of community engaged effort?

6. PLOS authors have the option to publish the peer review history of their article (what does this mean?). If published, this will include your full peer review and any attached files.

**Do you want your identity to be public for this peer review?** For information about this choice, including consent withdrawal, please see our Privacy Policy.

Reviewer #1: No

Reviewer #2: **Yes: **Sarah Cusworth Walker, PhD

---

## [Editor Report · Decision Letter 1]

Developing programme theory for a place-based, systems change approach to adolescent mental health: A developmental realist evaluation

PMEN-D-24-00539R1

Dear Dr Berry,

We are pleased to inform you that your manuscript 'Developing programme theory for a place-based, systems change approach to adolescent mental health: A developmental realist evaluation' has been provisionally accepted for publication in PLOS Mental Health.

Best regards,

Karli Montague-Cardoso

Executive Editor

PLOS Mental Health